# OpenReview forum: "Evaluating the Goal-Directedness of Large Language Models"
_ICLR.cc/2025/Conference — Submitted to ICLR 2025_

### Official Review · Reviewer_SAsD · 2024-10-17

**Soundness:** 2
**Presentation:** 1
**Contribution:** 2
**Rating:** 3
**Confidence:** 3

**Summary:**

This paper introduces a new benchmark to evaluate the "goal-directedness" of Large Language Models (LLMs); "goal-directedness" in this context is defined as "the propensity to use available resources and capabilities to achieve a given goal".  To do so, the authors create a BlocksWorld environment in which the "goal-directedness" deficit can be estimated as the difference between the expected rewards given that the capabilities are used to their fullest extend and the actual reward obtained . Then, the authors evaluate different LLMs in this environment (and ablations of it) demonstrating that current models lack the ability to use their resources and capabilities fully.

**Strengths:**

I want to preface this review by saying that LLMs and their evaluation is not my area of expertise so I might be not familiar enough with the current literature to assess the impact of the contributions of this paper. Overall I see potential in the area of research and the problem that is trying to be addressed. The problem is well motivated in terms of potential applications for the evaluation and understanding of LLMs.

1. **Neglectedness**: The study of goal directed behavior in LLMs is interesting, has some useful applications and has been almost completely neglected in current research directions.
2. **Safety and Ethics Implications**: The paper discusses the potential risks associated with more goal-directed AI agents, which is a valuable contribution to the state of AI safety and ethical considerations of AI.
2. **Ablation Study**: The ablation study does a good job at trying to isolate the components of the task and identify the most challenging aspects.

**Weaknesses:**

1. **Coherence of the metric and its goal**: My main issue with this paper is that the connection between the "goal-directedness" deficit and the actual metric that is introduced is not completely clear. More especifically, the authors state that they want to differentiate between two subtle cases: wether the model fails because it lacks the $\textit{capabilities}$ or the $\textit{motivation}$ to do the task. By definition,  $\text{goal-directedness-deficit(regret, capabilities)} = \mathbb{E}[\text{reward} | \text{optimal use of capabilities}] − \text{reward}$. This makes the assumption that the goal, namely stacking the blocks so as to minimize the height difference between two towers, is a task that is within the capabilities of LLMs. And this is likely to be false, as LLMs have been shown to be unable to perform well in seemingly simple tasks that require quantitative reasoning, e.g. reverse alphabetical sorting of words, even with chain of thought [1]. Therefore the metric fails to stablish the distinction on wether the evaluated LLMs fail because they lack capabilities or the motivation to do the task.

2. **Experimental rigor**: For all of the results plots it is not stated the number of evaluation seeds and the standard deviation of the  sample, making it impossible for the reviewers to make an assessment about the statistical properties of the result and validate some of the authors' claims (e.g. in line 412 the at"perhaps a statistical fluke"). More importantly, the authors derive their conclusions from a small number of datapoints (evaluation with only 3,4 and 5 blocks) and a single environment. Ideally a good benchmark should be able to show that the results might generalize to more complex scenarios, by showing results in a diversity of low complexity scenarios.
3. **Clarity**: Although the motivation is good, it was very difficult for me to understand what it was originally because some of its key ideas fail to be highlihted and reiterated (e.g. the importance of "goal-directedness" for safety and the distinction between "goal-directedness" and other reasoning tasks).

**References:**

[1] Suzgun, M., Scales, N., Schärli, N., Gehrmann, S., Tay, Y., Chung, H. W., Chowdhery, A., Le, Q. V., Chi, E. H., Zhou, D., & Wei, J. (2022). Challenging BIG-Bench tasks and whether chain-of-thought can solve them. arXiv preprint arXiv:2210.09261. https://arxiv.org/abs/2210.09261

**Questions:**

1. What was the reasoning for using a BlocksWorld environment as an indicator of "goal-directedness"?
2. Did you consider other potential tasks?
3. What would constitute a good task for the purpose of measuring goal-directed behavior, i.e. what properties were you looking for?
4. Why not fine-tune a model to provide more insight about the difficulty of the task?

---

> ### Author Response · Authors · 2024-11-19
>
> Thank you for your review of our paper, which you correctly describe as “demonstrating that current models lack the ability to use their resources and capabilities fully” and “well motivated in terms of potential applications for the evaluation and understanding of LLMs.”:
>
> Let us address the concerns that you raised:
>
> # Weakness 1: Do models have the necessary capabilities? If not, does the measurement mean much?
>
> It’s true that LLMs are far from universally capable AIs, and sometimes have surprising weaknesses and failure modes. (The paper you cite is already a few years old, so some of these failure modes have already disappeared.)
>
> For this reason, a significant part of our paper is devoted to assessing whether models do have the necessary capabilities to do the tasks that we test them on. In general they do. It’s also worth emphasizing that our tasks allow a range of outcomes beyond just fail/pass, and allow us to measure how close a model came to a “perfect” answer. The tasks are hard enough that no model solves them perfectly all the time, and easy enough for even GPT-3.5 to often make at least some progress on them.
>
> Furthermore, our goal-directedness metric is computed by comparing the performance of models on a combined task, with performance on different subtasks.
>
> We have also tailored the instructions, to minimize the frequency of models misunderstanding the task or interface.
>
> # Weakness 2. Seeds and results
>
> Apologies, the paper should of course mention this (we’ll make sure to bring it back). For the main results and ablations, models are always run on 5 seeds. The error bars show the Standard Error of the Mean (SEM). The standard deviation is about twice the width of the SEM bars (sqrt(5) times wider, to be precise).
>
> # Weakness 2: Small number of datapoints (evaluation with only 3,4 and 5 blocks) and a single environment
>
> What is a sufficient number of experiments is of course always somewhat subjective. But let’s us emphasize that within our blocksworld environment, we have tested models on a range of different tasks and objectives:
> * the Build Highest Tower task (which mostly requires measuring),
> * the Equal Towers task (which requires significant cognitive effort and execution skills), and
> * the Falling Tower task, which requires perseverance in spite of adversity.
> So even though it is a single environment, we still test a range of different kinds of tasks. And for each of these tasks, we have tested for a range of different numbers of blocks.
>
> Across essentially all these tasks and settings, the finding that most models lack some goal-directedness has been consistent.
>
> In reviewer yUcf’s opinion, “The experiments conducted are very close to "minimal but complete" in that they include all and only the aspects that are necessary to study goal-directedness, which makes for very clear results.”
>
> Since the submission of the paper, we have also developed an anagram environment. Please see the global comment.
>
> # Weakness 3, clarity
>
> Thanks for pointing this out. We will make sure to reiterate the importance of understanding goal-directedness also in the conclusion, and perhaps find a way to highlight it more in the introduction.
>
> As pointed out by reviewer yUcf, a nuanced understanding of LLMs is essential as LLMs become ever more widely deployed. In particular, many expect them to soon be deployed as “agents” acting autonomously on behalf of users, where goal-directedness is a key property for both safety concerns and capabilities.

---

> > ### Author Response · Authors · 2024-11-19
> > **Questions**
> >
> > # Question 1: Why blocksworld?
> > The blocksworld environment lets us formulate a range of different tasks and subtasks in a unified setting.
> >
> > # Question 2: Did you consider other potential tasks?
> > Yes, and after submission we also implemented an anagram environment. Other environments that we’ve been considering include agency evals frameworks such AgentBench [TODO: more?]. However, the blocksworld environment is simple and flexible, allowing us to analyze goal-directedness on various tasks.
> >
> > # Question 3: Criteria for task selection
> > A key aspect is that the environment should support tasks that can naturally be broken down into subtasks. As explained to reviewer yUcf, a breakdown needs to satisfy two key properties:
> > * It should be clear to the agent how to compose the subtasks to solve the full tasks. (This means we cannot break the task down into too miniscule components, such as steps of a Turing machine, since then it would not be obvious how to compose the tasks.)
> > * There should be no alternative way of solving a task other than the one suggested by the breakdown.
> >
> > The Equal Towers task naturally breaks down into a measuring phase, a planning phase, and an execution phase, none of which can realistically be avoided, and it’s sufficiently obvious to the agent that each of these phases have to happen.
> >
> > Beyond this, it is also important that the interface is clear and understandable to the agents, so that our results measure agent’s capabilities at the task, rather than their capabilities at understanding the interface (it took significant iterations to find an interface and environment that was palatable to all the agents that we wanted to test).
> >
> > Finally, the task must be suitably hard, so that models neither do it perfectly nor just random guessing. The blocks world in particular let us adjust the number of blocks, as an easy way to adjust task complexity
> >
> >
> > # Question 4: Why not fine-tune models?
> > Our goal is not to measure the difficulty of a task, but to assess the goal-directedness of any LLM agent. The framework we propose for doing this is general, flexible and can be applied to any model, including fine-tuned models. It would be an interesting line of future work, to see how far fine-tuning can improve the goal-directedness of LLM agents.
> >
> > Please let us know if our answers have addressed the concerns raised, or if you have additional questions.

---

> > > ### Comment · Reviewer_SAsD · 2024-11-23
> > >
> > > I thank the authors for replying to my own and other reviewer's concerns. I understand the definitions and the contribution of this work better after this discussion. However, I think that this is evidence that significant changes to the manuscript are necessary to communicate the benchmark protocol and its goal as clearly as possible. A more formal definition of the goal-directedness-gap, under the Reinforcement Learning framework could significantly enhance the presentation (from discussion with reviewer UoUh).
> > >
> > > Moreover, now that I better undertand the contribution I think that it is expected for any benchmark paper accepted at a top Machine Learning venue to have a review of related work, open source implementation and assesment of current SOTA models on said benchmark. This reduces the contributions by the authors to (1) a conceptual definition of goal-directedness suitable for LLMs, and (2) a principle for evaluating it empirically (from discussion with reviewer BDx1). These contributions do not seem to warrant a higher score.
> > >
> > > In my opinion a good benchmark should encourage further research on the area by having meaningful empirical applications (or by serving as a stepping stone to them) and it is still unclear to me that the benchmark could be useful under more realistic scenarios. As the authors have stated, measuring goal-directedness requires that the task can be broken down in subtasks, and that that there is no alternative way of solving the task other than the one suggested by the breakdown. This significantly limits the applicability of this benchmark (as a training or safety metric), especially without any results on the generalization properties of goal-directedness across LLMs and across tasks.
> > >
> > > Reviewer yUcf has also pointed out that there is a flaw in assuming access to the agent's capabilities since measuring those capabilities is indistiguishable from assesing the agent's overal performance in that specific sub-task, which might in fact require goal-directed behavior itself. Since, the breakdown is itself a sub-task the authors must then assume the granularity and partitioning of the sub-task breakdown. These two components make the overall benchmark seem very subjective and lacking theoretical motivation. I would be interested in understanding the author's perspectives regarding these points, especially if they feel that I may have misunderstood important elements of this work.

---

> > > > ### Comment · Reviewer_UoUh · 2024-11-24
> > > >
> > > > Appreciate the detailed responses by authors. I concurr with reviewer SAsD that the paper requires major revision and needs another round of review.

---

> > > > ### Author Response · Authors · 2024-11-26
> > > >
> > > > Thanks for your points, and glad to hear you feel like you understand it better now.
> > > >
> > > > _"A more formal definition of the goal-directedness-gap, under the Reinforcement Learning framework could significantly enhance the presentation (from discussion with reviewer UoUh)."_
> > > >
> > > > Fair point, we'll incorporate this.
> > > >
> > > > _"I think that it is expected for any benchmark paper accepted at a top Machine Learning venue to have a review of related work, open source implementation and assesment of current SOTA models on said benchmark"_
> > > >
> > > > We do have an extensive review of related work (Section 2). The implementation will be made open source with the release of the paper (line 197). We have evaluated both Gemini and several GPT models. We'll consider adding evaluations also for other models.
> > > >
> > > > _"This reduces the contributions by the authors to (1) a conceptual definition of goal-directedness suitable for LLMs, and (2) a principle for evaluating it empirically."_
> > > >
> > > > We agree that the conceptual definition and the principle for evaluating it empirically are key contributions. Goal-directedness is an important metric to keep track of as LLMs are increasingly deployed as agents. Therefore having a way to measure it is quite valuable.
> > > >
> > > > Another key finding is that current SOTA models do lack a fair bit of goal-directedness. This suggests that there is a "capability overhang" in that current models are not using their capabilities fully, which in turn suggests a path to making them more agentic,
> > > >
> > > > _"As the authors have stated, measuring goal-directedness requires that the task can be broken down in subtasks, and that that there is no alternative way of solving the task other than the one suggested by the breakdown. This significantly limits the applicability of this benchmark"_
> > > >
> > > > It's true that it's not possible to straightforwardly assess the goal-directedness of a model on any task. This is because assessing goal-directedness always requires disentangling motivation from capabilities, which is often non-trivial. However, what our paper shows is that it's still possible to assess goal-directedness of models by finding tasks that naturally break down into subtasks.
> > > >
> > > > While the goal-directedness of a model will likely vary somewhat between tasks, being able to assess it on a number of (decmoposable) tasks can still give an idea of how goal-directed a model is in general, which can guide further training strategies and safety mitigations.
> > > >
> > > > _"there is a flaw in assuming access to the agent's capabilities since measuring those capabilities is indistiguishable from assesing the agent's overal performance in that specific sub-task, which might in fact require goal-directed behavior itself. Since, the breakdown is itself a sub-task the authors must then assume the granularity and partitioning of the sub-task breakdown."_
> > > >
> > > > The assumption we're making here is that it requires less goal-directedness to do a smaller task than to do a larger task. This assumption seems strongly consistent with folk psychology in humans. For LLMs, it also seems quite easy to just answer a simple question (in fact, getting them not to do it is often the hard thing -- cf jail breaks), whereas larger tasks requiring chaining together many steps of reasoning will be harder for them. The larger tasks therefore require more "motivation", in some plausible generalisation of the term. Finally, it's also consistent with our empirical results.
> > > >
> > > > As we explained in our reply to yUcf, there will generally not be a breakdown of a task into some "primary" capabilities. Instead, there will typically be zero or more ways to break down a task into sub-tasks. Any particular breakdown can give us evidence of lack of motivation.

---

### Official Review · Reviewer_BDx1 · 2024-10-30

**Soundness:** 3
**Presentation:** 3
**Contribution:** 2
**Rating:** 3
**Confidence:** 4

**Summary:**

This paper investigates the performance of LLMs in goal-directedness behavior. It explores whether LLMs can effectively leverage their capabilities and resources to achieve defined objectives by defining goal-directedness, developing an evaluation framework, and assessing LLM performance in practical tasks.  The primary contribution of this paper lies in the design of the evaluation framework. This framework quantifies the extent to which agents utilize their capabilities toward a specific goal by measuring relevant agent abilities, predicting how agents would resolve problems if they fully employed these abilities, and comparing the predicted performance with the agents' actual problem-solving outcomes.

**Strengths:**

1. The goal-directedness approach enhances the effectiveness of LLM-based agents by providing robust constraints in terms of safety and ethics. Therefore, evaluating the goal-oriented nature of LLMs is of significant importance.
2. The clarity of the writing is commendable, facilitating comprehension.
3.  The design of the evaluation framework is intriguing, and the conclusions drawn regarding the goal-directedness  of several state-of-the-art LLMs are persuasive.

**Weaknesses:**

- The discussion of related work is insufficient. For instance, the paper "Can Large Language Models Reason About Goal-Oriented Tasks?" appears to provide a more comprehensive interpretation of the goal-oriented nature of LLMs. I recommend that the authors elaborate on the distinctions and connections between the two studies.
- The contributions of the paper are relatively limited, with the primary contribution seemingly revolving around the design of the goal-directedness deficit and the evaluation of several LLMs' performance.
- The experimental design is somewhat simplistic and does not adequately capture the goal-directedness of LLMs across more complex and diverse tasks. Further investigation is needed into the internal mechanisms of the models, specifically how decisions are made and how these decisions relate to their goal-directed behavior.
- It is essential to test the influence of auxiliary methods (e.g. COT, TOT) aimed at enhancing the reasoning capabilities of the models on the experimental outcomes.

**Questions:**

See weaknesses.

---

> ### Author Response · Authors · 2024-11-19
>
> Thank you for your review of our paper which measures the goal-directedness of LLMs, for which you found the “conclusions drawn regarding the goal-directedness of several state-of-the-art LLMs … persuasive.”
>
> Let us address each of the concerns you raised:
>
> # The discussion of related work is insufficient:
> Our paper provides an extensive literature review over related work both in machine learning and neuroscience and psychology, spanning more than a page and resulting in more than 3 pages of references.
>
> We will be happy to also include the paper you mention. Briefly, they are primarily interested in measuring the capabilities of LLMs to reason about goal-oriented tasks, while we are interested in the motivation of LLMs to use capabilities to solve such tasks.  A main finding of their paper is that Tree of Thoughts (ToT) reasoning is less effective for reasoning about sequential reasoning tasks, while Chain of Thought (CoT) prompting only works in certain sequential reasoning scenarios, and is detrimental in others. They also offer a useful characterisation of key properties of goal-directed tasks, which we’ll consider incorporating.
>
>
> # The contributions of the paper are relatively limited:
> As summarized at the end of our introduction, our contributions are five-fold:
> * a conceptual definition of goal-directedness suitable for LLMs, and
> * a principle for evaluating it empirically (Section 3),
> * an open-source implementation of 5 tests in a Blocksworld environment, and
> * an assessment of the goal-directedness of 4 LLMs (Section 4), as well as
> * a review of related work in psychology/neuroscience and AI (Section 2).
> As pointed out by reviewer yUcf, a nuanced understanding of LLMs is essential as LLMs become ever more widely deployed. In particular, many expect them to soon be deployed as “agents” acting autonomously on behalf of users, where goal-directedness is a key property for both safety concerns and capabilities.
>
> # The framework is too simplistic and would not capture goal-directedness in other tasks and domains?
>
> As reviewer yUcf points out, “The proposed definition of goal-directedness is clear, straightforward (this is a positive thing), and well-motivated. The experiments conducted are very close to "minimal but complete" in that they include all and only the aspects that are necessary to study goal-directedness, which makes for very clear results.”
>
> Furthermore, our framework can be easily used to evaluate LLM goal-directedness both during and after training, and the principle is easily transferable to other tasks and environments. The experimental design is carefully crafted to facilitate this. In our response to reviewer yUcf, we briefly describe how we have applied the same principle to an anagram environment.
>
> # Further investigation is needed into the internal mechanisms of the models.
>
> Mechanistic interpretability is beyond the scope of our work. Despite ongoing efforts to understand the internal workings of deep neural networks, and LLMs in particular, spanning 100s if not 1000s of papers from top universities and AI labs, the field is still far from answering much more basic questions such as how facts are stored or simple questions answered (the situation is not so different from Neuroscience, in fact). The fact that the behavioral approach pursued in our paper doesn’t rely on a detailed understanding of the internals of the models is one of its key strengths. We will emphasize this in the updated version of our manuscript.
>
> # It is essential to test the influence of auxiliary methods (e.g. COT, TOT):
>
> We effectively already test Chain-of-thought, as the system prompt encourages the models to “reason-step-by-step” before outputting their next action (line 183 in the paper). That said, we agree that it would be interesting to test the influence of various scaffolding techniques. We discuss this in the limitations section (line 482). We hope others will use our framework (which will be open sourced upon paper publication) to systematically explore how other scaffolding techniques impact the goal-directedness of models.
>
> Please let us know if our answer addresses your concerns or if you have additional questions.

---

### Official Review · Reviewer_yUcf · 2024-11-04

**Soundness:** 3
**Presentation:** 3
**Contribution:** 4
**Rating:** 6
**Confidence:** 4

**Summary:**

The paper proposes a mathematical definition of "goal-directedness" in the context of LLM-agents as the difference between the agent's actual and its expected performance on tasks given optimal use of its capabilities. The authors then introduce a toy Blocksworld environment and an associated task with three variants and proceed to measure the performance and goal-directedness of some LLM-agents on this task. The LLM-agents studied are based on GPT-3.5-turbo, GPT-4-1106, GPT-4o, and Gemini-1.5-Pro. They find that the agents differ in their capabilities and goal-directedness, and while they are generally all rather capable, they are not very goal-directed. The paper also discusses associated ablation experiments and closer analyses.

**Strengths:**

**Originality**

While other work has aimed to define goal-directedness in similar settings, the definition proposed in this paper seems to be novel, while also being elegantly simple and powerful in its generality.

**Quality**

The toy setting and task introduced in the paper are well-designed and highly appropriate for the investigation. The experiments are well-motivated, the ablations are interesting and useful, the analyses are informative and include confidence intervals. Overall, it is clear the authors had good attention to the question at hand and applied good intuition to what experiments would be most informative.

**Clarity**

The proposed definition of goal-directedness is clear, straightforward (this is a positive thing), and well-motivated. The experiments conducted are very close to "minimal but complete" in that they include all and only the aspects that are necessary to study goal-directedness, which makes for very clear results. The text flows well and the sections motivate each other. The plots are clear and informative.

**Significance**

As LLM-agents become increasingly widely studied and deployed, it becomes increasingly important to form nuanced understanding of them in general settings. This paper contributes to this. Goal-directedness as a trait of LLM-agents also has safety implications, and as such this contribution is all the more significant.

**Weaknesses:**

1. It is a significant issue that the calculation of goal-directedness assumes direct access to the agent's capabilities. This is because the measurement of the capabilities looks a lot like the measurement of the agent's overall performance on a task. Presumably it already will reflect the agent's goal-directedness in some capacity. Essentially, it creates a "turtles all the way down" problem. Breaking the task down seems like a reasonable thing to do, but not only is it imperfect in the sense above, it also requires some subjective judgement on the part of the experimenter (different experimenters can reasonably disagree on how a task should be broken down, especially in the case of more realistic tasks).

2. A concrete example of the issues above: prompting for motivation significantly affected the agent's motivation on the "Cognitive Effort" partial task. So the claim being made here is that this partial task is already very "goal-directedness"-loaded, i.e. "goal-directedness" is a significant component of the agent's performance on the task. But this is already a broken-down task, and to get to direct capabilities, the authors only break it down further into two component capabilities. How can we be confident that we have now reached a pure source for capabilities measurements?

3. The only results are shown in the toy BlocksWorld environment. I think it makes sense for the main experiments to be conducted there, but I am left very curious to see how the framework can be applied to realistic tasks, and in particular, whether the results would hold there.

I would like to note that while I maintain that the above are significant issues, I still feel positively about the paper's contribution, and hope to see further work in this direction making progress on the issues raised.

**Questions:**

1. Do the authors agree with the issue I point out in Weakness 1? How do they think the presumed purity of the capabilities measurements can be justified?

2. Similarly, how do the authors recommend that a task be broken down into its constituent capabilities? Is there an objective general procedure?

3. Is goal-directedness here claimed to be a property of an agent? Is there any claim as to how the goal-directedness measurements on an agent would differ across different tasks? We do expect the same agent to be differently capable at different tasks. Should they also be differently goal-directed? From the conceptual motivation, I would hope this is not strongly the case, but I expect that experimental results might show otherwise. Do the authors agree?

---

> ### Author Response · Authors · 2024-11-19
>
> Thank you for your feedback and your positive review!
>
> # Weakness 1: "turtles all the way down"?:
>
> This is a good point. There is in general no single, objective way to break down a task. Nevertheless, any particular breakdown can still give evidence of a lack of goal-directedness, if the performance on the combined task is worse than would be expected from the performance on the subtasks. This relies on a tacit assumption that doing smaller, more well-scoped tasks requires less goal-directedness than doing larger tasks.
>
> A breakdown needs to satisfy two key properties:
> * It should be clear to the agent how to compose the subtasks to solve the full tasks. (This means we cannot break the task down into too miniscule components, such as steps of a Turing machine, since then it would not be obvious how to compose the tasks.)
> * There should be no alternative way of solving a task other than the one suggested by the breakdown.
>
> We will endeavor to highlight these important points more in the updated manuscript.
>
> # Weakness 2. How can we be confident that we have now reached a pure source for capabilities measurements?
>
> We’re not confident that there usually is such a thing as a complete breakdown of a task into “primary” capabilities. As discussed in the response to Weakness 1 above, there will typically be many ways to break down a task into smaller capabilities. Any breakdown that satisfies the principles above can give evidence about lack of goal-directedness.
>
> # Weakness 3. The only results are shown in the BlocksWorld environment.
>
> The reason we choose the blocksworld environment is that it lets us formulate a range of different kinds of tasks and objectives in a unified environment. Thus we were able to define:
> * the Build Highest Tower task (which mostly requires measuring),
> * the Equal Towers task (which requires significant cognitive effort and execution skills), and
> * the Falling Tower task, which requires perseverance in spite of adversity.
> So even though it is a single environment, we still test a range of different kinds of tasks.
>
> Nevertheless, we do recognise that testing the principle in other environments would also be valuable. Please see the global comment describing another setting in which we’ve tested our goal-directedness metric.
>
> # Questions 1 and 2
>
> See responses to weakness.
>
> # Question 3
>
> Goal-directedness is a property of an agent and may well vary across tasks -- different tasks have different levels of complexity and require the use of different capabilities. Nevertheless, the fact that both Build Equal Towers and Falling Towers tasks, as well as the just implemented anagram task, have the same “winner” (Gemini-1.5-Pro), indicates there may be some level of generalization across tasks.

---

> > ### Comment · Reviewer_yUcf · 2024-11-26
> >
> > I thank the authors for the helpful response, as well as the other reviewers for the thoughtful comments.
> >
> > I still feel positively about this work, but perhaps less strongly than initially. After more careful thought, I'm coming to view the weakness 1 and question 3 from my initial review as being more significant issues than I initially realized. I now also agree with
> > reviewer BDx1 comments that the contributions of the paper are relatively limited. Although I still commend the quality and clarity of the experiments that were presented, the bottom line is perhaps insufficiently significant.
> >
> > I see two main paths by which this work could be more clearly significant: either the measure of goal-directedness could be shown to have "nicer" and more robust properties (e.g. maybe there is a principled algorithm for breaking down tasks into component capabilities -- this is related to my weakness 1 --, or maybe goal-directedness of a given agent is shown to be consistent across tasks requiring different capabilities -- this is related to my question 3), or the distinction between goal-directedness and capabilities could be shown to be more "useful" (e.g. it facilitates clear and generalizable interventions -- such as the intervention of telling the agents to try harder, which the authors tested, except more powerful or more systematically investigated).
> >
> > I remain excited to see further work by the authors, but for now I downgrade my decision recommendation from "accept" to "marginal accept".

---

> > > ### Author Response · Authors · 2024-11-27
> > >
> > > Thanks for your suggestions!
> > >
> > > > either the measure of goal-directedness could be shown to have "nicer" and more robust properties (e.g. maybe there is a principled algorithm for breaking down tasks into component capabilities -- this is related to my weakness 1
> > >
> > > Any measure of goal-directedness will rely on a way of disentangling the agent's capabilities from its motivation. This is feasible for tasks for which there is a clear set of required capabilities, and a single natural way to solve the task given those capabilities. I think we can do a better job at articulating this principle, and showing how it can apply across different contexts.
> > >
> > > We're not sure it's important that it can be done for *all* tasks, as it seems interesting enough if we can establish a lack of goal-directedness across a reasonable range of tasks.
> > >
> > > > or maybe goal-directedness of a given agent is shown to be consistent across tasks requiring different capabilities -- this is related to my question 3)
> > >
> > > Does the anagram environment described in our top comment help address the concern of BlocksWorld specificity? https://openreview.net/forum?id=BECkhjcofz&noteId=eCcPdjHkKw
> > >
> > > > the distinction between goal-directedness and capabilities could be shown to be more "useful" (e.g. it facilitates clear and generalizable interventions -- such as the intervention of telling the agents to try harder, which the authors tested, except more powerful or more systematically investigated).
> > >
> > > This is a nice idea, and something we will look further into!

---

### Official Review · Reviewer_UoUh · 2024-11-04

**Soundness:** 2
**Presentation:** 2
**Contribution:** 2
**Rating:** 3
**Confidence:** 4

**Summary:**

Authors defined the concept of goal-directedness for LLM-based agents and measured this metric in tasks that involve information gathering, planning, and execution phases. They empirically probed four SOTA LLM agents to tackle a block stacking task focusing on different combinations of the phases. They found limited goal-directedness across all LLMs with Gemini performing better overall.

**Strengths:**

* New Metric: There is a huge interest in the community to better evaluate LLMs for planning and the goal-directedness can play a role

* Empirical evaluation: Authors highlighted that current LLMs, despite their sophisticated capabilities, do not fully deploy these capabilities toward goal achievement.

* It was refreshing to see authors focused on stochastic environments.

**Weaknesses:**

* Readability: The paper is hard to follow and the definitions are not rigourous. In 79, you mentioned "The gap between actual and expected performance given optimal use of capabilities". To me this means the difference between expected return under optimal policy and observed return under optimal policy. Not sure how this translates to goal-directedness as it only captures domain stochasticity. In 171, you mean expected return vs. expected reward. Reward is instantaneous while return is the cumulative return. In 174, can you define E[regret | optimal use of capabilities]? The second eqn. means how much the return could improve with additional capabilities. I don't think it is the same as goal-directedness-deficit.


* Generalization Issues (Line 139): The statement that LLMs do not ask clarifying questions might not fully apply, as some LLMs, like GPT-4, demonstrate this capability.

* Terminology Ambiguity (Lines 159 & 171): Concepts such as “capability-conditioned goal-directedness” and expected “return” versus “reward” lack clarity and could be better explained.

* Undefined Terms (Lines 174, 289 & 313): Key terms, including
E[regret∣optimal use of capabilities], "unexplained_regret," and "statistical sophistication," need explicit definitions to improve comprehensibility.

* Algorithm Motivation (Line 244): Lacks clear motivation for Algorithm 1. Would be great to discuss its creation.

* Inconsistent Noise Parameters (Line 250): On 205 you mentioned sigma to be .1 * true height, yet here you also added a constant factor.

* Presentation Quality (Various Lines): Figures and images are of low quality, lacking necessary labels (e.g., missing legend in Fig 8.b) and captions (e.g., Fig 4), and should be vectorized with larger fonts for readability.

Minor:
* 389: "Gemini performs better than would be expected from" Fix grammar
* 492: Whence -> hence
* 504: language-based LLM -> LLM as L stands for language

**Questions:**

* (Line 369): Why does the regret for GPT 3.5 fluctuate (i.e., drop and then increase) in the observed experiments?

---

> ### Author Response · Authors · 2024-11-19
>
> Thank you for your review, which recognises we found limited goal-directedness in state-of-the-art language models, using a method reviewer yUcf described as “novel, while also … elegantly simple and powerful in its generality.”
>
> # Rigorous definitions
>
> _Does Line 79 mean the difference between difference for the optimal policy?_
>
> No, and apologies the paper was insufficiently clear on this point. The comparison is not intended to involve the optimal policy at all (only optimal use of capabilities). We will change the misleading sentence on line 79 to "The gap between the agent's actual performance and the agent's expected performance if it made full use of its capabilities."
>
> For example, the Build Highest Tower task requires the agent to build a maximally high tower out of any two blocks. The optimal policy takes measurements until all block heights are known to two decimals with high certainty, then stack the two highest blocks. But agents typically only have the capability (or patience) to measure block heights to one decimal point. Conditioning on this (lack of) capability, an agent’s expected performance will be worse than optimal (as it won’t be able to reliably select the highest block).
>
> However, it often turns out that agents’ suboptimal performance is not fully explained by their lack of capabilities. Instead, they perform even worse than one would expect from just knowing their capability to measure block heights accurately (and ability to select the blocks with the highest estimates). The gap between this actual performance, and the expected performance given full use of capabilities, is what we interpret as a goal-directedness deficit. Figure 5 in the paper shows this gap for the build highest tower task (and Section 4.3 discusses it).
>
> _Formal def of E[regret | optimal use of capabilities]? (Line 174)_
>
> Formally in RL terminology, the agent is inhabiting a POMDP, where its capabilities correspond to options. For example, an agent capable of measuring blocks to 1 decimal can be viewed to have a measuring option which brings it to an (information) state where the block height is known to 1 decimal. The capability-options define an MDP over states and belief states, with an action-set consisting of these capability-options.
>
> Now we can more formally define:
>
> E[regret | optimal use of capabilities] = optimal return in the original task - the return of the optimal policy in the capability-options belief-MDP.
>
> # Algorithm 1 motivation
>
> The idea behind Algorithm 1 is simply to simulate an agent with a particular set of capabilities.
>
> To this end, we first sample a set of “actual” block heights (Step 2).
>
> We then account for the agent’s measurement capability, by sampling a measurement error for the height of each block, using the distribution of errors we’ve seen the agent make in a separate capability test (Step 3).
>
> Next, we assess the agent’s planning ability, by
> * Sampling a set of configurations that the agent might have conceived of, based on the number of configurations the agent generated in a separate capability test (Step 4),
> * Computing the height the agent would have assigned to each configurations, according to its flawed block heights estimates from Step 3 above (Step 5), and then
> * Picking a preferred configuration based on these height estimates, here taking into account the agent’s ability at picking out the best configuration given some height estimates (Step 5), as usual based on an independent assessment of the agent’s ability at this
>
> Finally, we take into the agent’s ability at actually building the tower it intends to (Step 8). Again, this is based on an independent assessment of the agent’s ability to execute an intended plan.
>
> _Line 250: On 205 you mentioned sigma to be .1 * true height, yet here you also added a constant factor._
>
> Thanks for pointing this out. We added a constant term to our model of agent measurement errors, to account for the possibility that agents adapt the number of questions to the noise observed in the measurements. An explanation would have been warranted.
>
> On reflection, we have decided that it is cleaner to instead just sample one of the agent’s actual measurement errors on a block of similar height from the capability check. This sidesteps the need to fit a model, and the extra assumptions this entails. The results are only marginally affected. We will update the paper to reflect this change.
>
> # Result analysis
>
> _Why does the regret for GPT 3.5 fluctuate (i.e., drop and then increase) in the observed experiments?_
>
> The slight dip at 4 blocks may well be a statistical fluke, as the mean at 5 blocks is well within the SEM of the 4-block result. More surprising perhaps is the dip at 4 blocks for both GPT-3.5 and GPT-4 on line 420. One possible explanation for this is that putting an equal number of blocks in each tower is a reasonable heuristic for making equally high towers. This heuristic works better for an even number of blocks.

---

> > ### Author Response · Authors · 2024-11-19
> > **Minor points**
> >
> > Line 171: Apologies, we used the term reward informally here. We will clarify this in an updated version, either switching to return (as you suggest), or using ‘loss’ for a broader ML interpretation (in either case, specifying precisely what we mean with the term).
> >
> > Line 139: We will update the wording to “Their lack of goal-directedness is further evidenced by some models not asking clarifying questions.”
> >
> > Line 159: We will clarify the sentence “In contrast, our work measures capability-conditioned goal-directedness, taking into account the agent’s (lack of) capabilities when assessing its goal-directedness.” to make this paragraph better understandable in isolation.
> >
> > Line 289: unexplained_regret is the same as goal-directedness deficit. We will remove the term unexplained_regret and only use goal-directedness deficit throughout. Apologies for the confusion.
> >
> > Line 313: What we mean by “Statistical sophistication” is explained in the subsequent sentence, i.e. it’s the propensity to adapt the number of measurements based on how noisy they are. Please let us know if you think further clarification would be useful here.
> >
> > missing legend in Fig 8.b and caption Fig 4. Apologies. The legend in Fig 8b is the same as for most of the other plots in the paper, but we will add this back to this plot for clarity (especially since the legend beside has a different legend). We will expand the Fig 4 caption, and increase the font size for the axes.
> >
> > Typos: Thanks, fixed.

---

### Author Response · Authors · 2024-11-19
**Anagram environment**

Multiple reviewers have asked how the results generalize beyond the blocksworld environment in which our main evaluations are carried out.

Since the submission of the paper, we have developed an anagram environment, where the agent gets a sequence of letters, and is asked to permute them into one or more valid English words. This is a common setting for assessing human goal-directedness.

We compare actual performance on this task, with the agent’s expected performance based on its capabilities to
* permute a sequence of letters in many different ways, and to
* recognize whether an arbitrary permutation is an English word or not.

We find that the lack of goal-directedness is less significant in this setting, but that several models still demonstrate a lack of goal-directedness, especially for short sequences of letters (2 or 3). For longer sequences, models struggle to generate all or even most permutations of the letters (the number of permutations increases exponentially), meaning that their expected performance is sometimes worse than their actual performance, suggesting that the models have an alternative way of finding anagrams that doesn’t involve explicitly generating and checking permutations of the letters.

As for the task we already considered in the paper, Gemini comes out as the most goal-directed model also on the anagram task.

---

### Meta-Review · Area_Chair_Ag1r · 2024-12-19

**Metareview:**

This work defines the concept of goal-directedness for LLM agents, and develops a framework for evaluating it empirically on tasks involving information gathering, information processing, and execution. Although some reviewers expressed interest in the research question and proposed metrics, considering them novel, multiple reviewers also pointed out issues with experimental rigor, unclear writing, and limited task diversity. AC encourages the authors to refine their work and extend the experiments to a broader range of tasks and settings, believing it could become an appealing contribution.

**Additional Comments On Reviewer Discussion:**

The final score for this paper is 6333, with a reduction from 7 to 6. Two reviewers responded to the author. The results indicate that the author did not fully address the reviewers' concerns, and one of the reviewers believed that the paper required major revisions and a re-evaluation after the rebuttal.

---

### Decision · Program_Chairs · 2025-01-22

Reject